# The Anti-Inflammatory Effects and Mechanism of the Submerged Culture of *Ophiocordyceps sinensis* and Its Possible Active Compounds

**DOI:** 10.3390/jof10080523

**Published:** 2024-07-27

**Authors:** Hsien-Chi Huang, Yu-Juan Shi, Thuy-Lan-Thi Vo, Tai-Hao Hsu, Tuzz-Ying Song

**Affiliations:** 1PhD Program of Biotechnology and Bioindustry, College of Biotechnology and Bioresources, Da-Yeh University, Changhua 515, Taiwan; d0367004@cloud.dyu.edu.tw; 2Department of Medicinal Botanicals and Foods on Health Applications, Da-Yeh University, Changhua 515, Taiwan; b0970705060@gmail.com (Y.-J.S.); d0467601@cloud.dyu.edu.tw (T.-L.-T.V.); th0123@mail.dyu.edu.tw (T.-H.H.)

**Keywords:** *Ophiocordyceps sinensis*, microglial, anti-inflammatory, cytokines, submerged culture, neurodegenerative diseases

## Abstract

The pharmacological effects of the fruiting body of *Ophiocordyceps sinensis* (*O. sinensis*) such as antioxidant, anti-virus, and immunomodulatory activities have already been described, whereas the anti-inflammatory effects and active components of the submerged culture of *O. sinesis* (SCOS) still need to be further verified. This study aimed to investigate the active compounds in the fermented liquid (FLOS), hot water (WEOS), and 50–95% (EEOS-50, EEOS-95) ethanol extracts of SCOS and their anti-inflammatory effects and potential mechanisms in lipopolysaccharide (LPS)-stimulated microglial BV2 cells. The results demonstrated that all of the SCOS extracts could inhibit NO production in BV2 cells. EEOS-95 exhibited the strongest inhibitory effects (71% inhibitory ability at 500 µg/mL), and its ergosterol, γ-aminobutyric acid (GABA), total phenolic, and total flavonoid contents were significantly higher than those of the other extracts (18.60, 18.60, 2.28, and 2.14 mg/g, *p* < 0.05, respectively). EEOS-95 also has a strong inhibitory ability against IL-6, IL-1β, and TNF-α with an IC_50_ of 617, 277, and 507 µg/mL, respectively, which is higher than that of 1 mM melatonin. The anti-inflammatory mechanism of EEOS-95 seems to be associated with the up-regulation of PPAR-γ/Nrf-2/HO-1 antioxidant-related expression and the down-regulation of NF-κB/COX-2/iNOS pro-inflammatory expression signaling. In summary, we demonstrated that EEOS-95 exhibits neuroinflammation-mediated neurodegenerative disorder activities in LPS-induced inflammation in brain microglial cells.

## 1. Introduction

The caterpillar fungus *O. sinensis* is known as Yarshagumba in Nepal and Dong-chong-xia-cao in Chinese, and it has also been found in the alpine regions of Bhutan, India, China, and Nepal [1,2]. It has been treasured for centuries in traditional medicine [2]. Primarily used as a tonic to boost immunity, it holds immense potential for various therapeutic applications [3]. Several studies have documented its diverse beneficial properties, making it a fascinating subject for scientific exploration [4]. Previous research has highlighted the potential of *O. sinensis* in addressing a multitude of health concerns [1,2,3,4]. Many pharmacological effects have been reported, including on the nervous system, immune regulation, and renal, liver, and cardiovascular diseases; in particular, it was also used to treat the severe acute respiratory syndrome virus (SARS virus) 2003 in China [3,4]. In addition, Sen and co-authors comprehensively pointed to phytochemicals such as phenolic acids, amino acids, fatty acids, sterols, polysaccharides, nucleosides, etc., which have contributed various beneficial nutritional or pharmacological properties [4]. Traditionally, *O. sinensis* has been consumed in powdered form, mixed with milk or water, or added to tea or soup. The high price of *O. sinensis* is mainly due to its medicinal value, which has led to an increasing demand from consumers and a gradual decrease in the production of natural wild species (fruiting bodies). Wei et al. warn that climate change further threatens its distribution in the wild [5]. Based on the above reasons, some researchers have used mycelium isolation technology and isolated *O. sinensis* mycelia for submerged culture [3]. The benefits of this new approach are twofold: it reduces pressure on wild populations and promotes ecological sustainability while providing a more economically viable source of *O. sinensis.* Although many studies were conducted in the past to demonstrate the effects of the fruiting body of *O. sinensis* on human health, the functionality of submerged cultures of *O. sinensis* mycelia and the active compounds in mycelia and fermentation still need to be further verified.

Neurodegenerative diseases (NDs) are a group of progressive conditions that affect millions of people worldwide, predominantly in their later years. These devastating disorders attack the nervous system, particularly the brain, leading to a decline in cognitive and motor function. The rising number of ND cases puts immense pressure on healthcare systems, families, and communities [6,7,8,9,10]. While the specific causes of each ND vary, researchers have identified several key factors that contribute to their development; these include oxidative stress (an imbalance between free radicals and antioxidants) leading to cellular damage, protein aggregation (the abnormal buildup of misfolded proteins in the brain, forming toxic clumps), neuro-inflammation (chronic inflammation in the nervous system damaging brain cells), neurotransmission impairment (disruption of chemical communication between neurons), mitochondrial dysfunction (impaired energy production in brain cells), and excitotoxicity (excessive stimulation of brain cells, leading to their death) [11].

Neuroinflammation is a defensive response of the brain to injury or infection, occurring in both chronic and acute neurodegenerative disorders. Microglia, the innate immune cells of the central nervous system, become rapidly activated when the brain is infected, injured, or damaged. They regulate the production of proinflammatory cytokines and immune response mediators, including nitric oxide (NO), tumor necrosis factor-α (TNF-α), interleukin-1β (IL-1β), reactive oxygen species (ROS), and various neurotoxic mediators [12,13]. Therefore, inhibiting the activation of microglia and neuroinflammation is a potential therapeutic target for reducing ND and neuronal damage.

The important bioactive compounds detected in Os mycelia are as follows: adenosine, polysaccharides, ergosterol, cordycepin, etc. [14,15]. These compounds have been reported to have various biological and pharmacological effects such as immunomodulatory, anti-inflammatory, and antioxidant effects [3]. Li et al. reported that the bioactive compounds of mycelia isolated from wild *O. sinensis* had neuroprotective effects on PC12 cells and prevented hydrogen peroxide-induced nerve damage in rats [16]. However, to date, no analytical data on the effects of SCOS on BV2 microglia or analysis of the bioactive components of their mycelial extracts are available.

Thus, in the present report, we explored the contents of active compounds of fermented liquid (FLOS), and various mycelial extracts (hot water—WEOS; 50% ethanol extracts**—**EEOS-50; and 95% ethanol extracts**—**EEOS-95) of SCOS by RP-HPLC assays and evaluated their potential cytotoxicity and role as a mediator of nitric oxide (NO) production. EEOS-95 (a potent extract) was then evaluated for its anti**-**inflammatory activities (cytokine release; interleukins-1β (IL**-**1β), interleukins-6 (IL**-**6), tumor necrosis factor**-**α (TNF**-**α), and prostaglandin E2 (PGE2)) in LPS**-**stimulated microglial BV2 cells through ELISA kits (LPS stimulation can mimic the initial acute inflammatory response to produce cytokines). Additionally, the effect of EEOS-95 on inflammatory proteins (inducible NO synthase (iNOS), cyclooxygenase**-**2 (COX**-**2), and nuclear factor kappa B (NF**-**κB)) and anti**-**inflammatory proteins (nuclear factor**-**erythroid 2**-**related factor 2 (Nrf**-**2), peroxisome proliferator**-**activated receptors (PPAR**-**γ), and heme oxygenase**-**1 (HO**-**1)) were determined in LPS**-**stimulated microglial BV2 cells by using Western blotting assays.

## 2. Materials and Methods

### 2.1. Chemicals

The mouse microglial BV2 cell line was purchased from the Food Industry Research and Development Institute (Hsinchu, Taiwan). The lipopolysaccharide (LPS; *Escherichia coli* O26:B6) and bovine serum albumin were bought from Sigma-Aldrich Co. (St Louis, MO, USA). RPMI 1640, and fetal bovine serum (FBS) were purchased from Gibco by Life Technologies (Frederick, MD, USA). All solvents were of HPLC grade, and the purity of all standard powders was > 99%. Tips, dishes, test tubes, etc., for cell culture were bought from Thermo Fisher Scientific (Waltham, MA, USA).

### 2.2. Preparation of Hot Water and Ethanol Extracts from the Submerged Culture of O. Sinesis (SCOS)

The *O. sinensis* H101 strain was isolated from wild *O. sinensis* (Qinghai Province, China), and identified by the Bioresource Collection and Research Center (Hsinchu, Taiwan) (BCRC 930166). The submerged culture of fungus contains its submerged mycelia and culture liquid is determined as a liquid culture of the dried mycelia under laboratory conditions. This experiment used dried mycelia (*O. sinensis* H101), kindly provided by Professor Hsu Tai-Hao. *O. sinensis* H101 was cultured at pH 7.0, in 100× *g*, and at 18 °C for 15 days in a 2 L Erlenmeyer flask containing (3% glucose, 0.5% peptone, 0.3% yeast extract, 0.1% H_2_PO_4_, and 0.05% MgSO_4_.7 H_2_O) on a rotary shaker (120 rpm). The submerged culture of Os H101 (SCOS) was harvested at the end of the fermentation process. Mycelia of SCOS were extracted with a 1:20 ratio (*w*/*v*) of 121 ± 2 ◦C hot water (in an autoclave) for 15 min, or extracted with a 1:20 ratio of 50% and 95% ethanol by soaking at room temperature for 24 h. The extracts were centrifuged (1500 rpm, 10 min) and concentrated at reduced pressure using a rotary evaporator. Then, the mycelia of the SCOS extracts were freeze-dried (lyophilization) to obtain the hot water extract of *O. sinensis* (WEOS), 50% ethanol extract of *O. sinensis* (EEOS-50), and 95% ethanol extract of *O. sinensis* (EEOS-95). Finally, samples were stored at 4 °C for the following analysis. The extraction yields of WEOS (32.9%), EEOS-50 (25.1%), and EEOS-95 (7.4%) were determined. The fermented liquid of Os (FLOS) was concentrated 10 times by using a rotary evaporator and then freeze-dried to powder. The preparation procedure was performed according to our previous study and Javadi et al., 2021 [17,18,19].

### 2.3. Determination of Bioactive Compounds

The determination of the bioactive compounds of SCOS was carried out using an Agilent 1200 reversed-phase high-performance liquid chromatograph coupled with a diode-array detector (Hitachi, Chiyoda City, Japan, Chromaster 5430). A HIQ Sil C18W reversed-phase column was used (4.6 mm × 250 mm, 5 µm). The results were expressed in mg/g. All solvents were of HPLC grade and filtered before HPLC analysis.

#### 2.3.1. Measurement of Adenosine

Adenosine was measured by applying the method described in Chang et al. [20]. The sample or standard was dissolved in 1 mL distilled water, and ultrasound-assisted (ultrasonic power 100 W) extraction was carried out for 1 h. The mobile phase was CH_3_OH:0.02 M KH_2_PO_4_ (15:85, *v*/*v*), with isocratic elution. The flow rate was 0.7 mL/min, the absorbance was measured at 254 nm, and the injection volume was 20 µL.

#### 2.3.2. Measurement of Ergosterol

Ergosterol was measured as described by Yuan et al. [21]. The sample or standard (10 mg) was dissolved in 1 mL methanol/dichloromethane (75:25, *v*/*v*) and shaken for 1 h. The mobile phase consisted of solvents A (80% methanol) and B (75% methanol in dichloromethane) with a gradient elution as follows: 0–5 min (100% A); 5–19 min (0%–100% B); 20 min (100% B); and 34–35 min (100% B). The flow rate was 1.0 mL/min, the absorbance was measured at 280 nm, and the injection volume was 20 µL.

#### 2.3.3. Measurement of GABA

The measurement of GABA was performed as previously described by Rogério da Silva Moraes et al. [22]. The mobile phase was A: 50 mM sodium acetate, 5% methanol, 2-propanol (pH 5.67); B: 70% methanol. The gradient was as follows: 0–10 min: 100% A, 10–20 min: 70% A, 20–25 min: 50% A, and 25 min: 100% A. The flow rate was 0.8 mL/min, the absorbance was measured at 338 nm, and the injection volume was 20 µL.

The chemicals were prepared as follows: the working solution was derivatized (OPA-NAC complex) to allow for a reaction between the OPA and NAC thiol group: 16.3 mg N-acetylcysteine (NAC), 13 mg o-phthalaldehyde (OPA), and 300 µL methanol were allowed to react for 30 min at room temperature in the dark, and were then stored in plastic test tubes at 4 °C in the dark (used within 24 h after preparation) with borate buffer (pH 9.6).

GABA standard or samples were mixed: 60 µL standard or sample, 40 µL borate buffer, and 10 µL OPA-NAC complex; after 10 min of reaction, this final solution was vortexed before HPLC analysis.

#### 2.3.4. Measurement of EPS, TPC, and TFC Contents

Extracellular polysaccharides (EPSs) were detected following the phenol–sulfuric colorimetric method described by Jiménez et al. [23].

The total phenolic content (TPC) of each extract was determined by applying the Folin–Ciocalteu method described by Yang et al. [18]. TPC was expressed as milligrams of gallic acid equivalent per gram of SCOS extract (mg Ga/g). The total flavonoid content (TFC) of each extract was determined using the aluminum chloride colorimetric method described by Yang et al. [18]. TFC was expressed as milligram of quercetin equivalent per gram of SCOS extract (mg Que/g).

### 2.4. Cytotoxicity Test of SCOS on BV2 Microglial Cells

The mouse microglial BV2 cell line was purchased from the Food Industry Research and Development Institute (Hsinchu, Taiwan). Cells were cultured in RPMI 1640, containing 2 mM L-glutamine, 1.5 g/L sodium bicarbonate, 10% fetal bovine serum, and 1% antibiotic penicillin/streptomycin. The medium for cells used in the experiments was changed every 2 days and incubated at 37 °C and 5% CO_2_.

The cytotoxicity evaluation of SCOS on BV2 microglial cells was determined by the MTT method described in Vo et al. [19]. Cells were treated with various concentrations (10–1000 µg/mL) of FLOS, WEOS, EEOS-50, and EEOS-95 and cell viability was detected at 490 nm using an ELISA reader (Synergy HTX, BioTek, Winooski, VT, USA).

### 2.5. Measurement of Nitrite Production

The evaluation of the effect of SCOS on NO level was applied by measuring nitrite accumulation as described in Vo et al. [19] with minor modifications. BV2 cells (5 × 10^5^ cells/mL) were seeded in 24-well plates for 2 h. Cells were treated with 10–1000 µg/mL of FLOS, WEOS, EEOS-50, and EEOS-95 for 24 h. They were LPS stimulated (1 µg/mL) for 24 h, and nitrite levels were detected at 540 nm using an ELISA reader (Synergy HTX, BioTek, Winooski, VT, USA). The inhibition % formula was: $Inhibition\;\left(\%\right)=\left(\frac{LPS-Sample}{LPS-CON}\right)\times 100$.

### 2.6. Anti-Inflammatory Activities of 95% Ethanol Extract SCOS (EEOS-95)

#### 2.6.1. Measurement of Pro-Inflammatory Cytokine Level

BV2 cells (5 × 10^5^ cells/well) were seeded in 24-well plates for 2 h and treated with different concentrations (50–500 µg/mL) of EEOS-95 for 24 h, and then incubated for 24 h with 1 µg/mL LPS to induce the secretion of inflammatory-related cytokines. After incubation, the concentration of cytokines in the culture medium was determined using an ELISA kit according to the manufacturer’s instructions. Cytokines were measured as IL-1β, IL-6, TNF-α (Invitrogen Co. Camarillo, CA, USA), and PGE_2_ (Life Technologies Corp. Frederick, MD, USA).

#### 2.6.2. Western Blotting Analysis

Cell pellets were harvested and washed with cold phosphate buffer saline (pH 7.4). An amount of 50 µg of cytoplasmic proteins were electrophoresed on 10% sodium dodecyl sulfate–polyacrylamide gels (SDS-PAGE) and transferred to a poly-vinylidene fluoride membrane. The membrane was blotted and incubated with specific primary antibodies overnight at 4 °C, followed by incubation with a horseradish peroxidase-conjugated secondary antibody. Finally, the blots were probed using enhanced chemiluminescence and autoradiographed. The relative density of protein expression was quantified using ImageJ software v1.8.0, developed by Wayne Rasband at the National Institutes of Health and the Laboratory for Optical and Computational Instrumentation (LOCI, University of Wisconsin, Madison, WI, USA). The protein content in the supernatant was determined using the BCA protein assay kit (Thermo, Rockford, IL, USA). Western blot analysis detected the protein expression of iNOS, COX-2, NF-κB, HO-1, Nrf-2, PARP-γ, and β-actin in the BV2 microglial cells.

### 2.7. Statistical Analysis

All statistical analyses were performed using SPSS for Windows, version 18 (SPSS Inc., Chicago, IL, USA). Data are expressed as means ± standard deviation and analyzed using one-way ANOVA followed by Duncan’s multiple range test. *p* < 0.05 is considered statistically significant.

## 3. Results

### 3.1. Bioactive Compounds in Extract of SCOS

We prepared the dry matter of the fermented liquid (FLOS) and the mycelial extracts with hot water (WEOS), 50% ethanol(EEOS-50), and 95% ethanol (EEOS-95) from the submerged culture of Os (SCOS). The following Bioactive compounds, total polyphenols, total flavonoids, adenosine, ergosterol, polysaccharide, and GABA, in SCOS were detected by RP-HPLC and colorimetric methods (Table 1). Instrument calibrations in analysis methods using liquid chromatography (LC) are usually created using either average correlation coefficients (r) or linear regression equations. The relative standard deviation (RSD%) was applied to evaluate the reproducibility of the chemical analysis. As shown in Table 1, the standard chemicals had an RSD% less than 101%. All of these results suggested that the analytical method could be applied to six active compounds’ analysis with excellent repeatability and stability [24]. The HPLC chromatograms of adenosine, ergosterol, and GABA standards and EEOS-95 are shown in Figure 1A, B, C, and D, respectively. The spectrum of the full wavelength of the sample (Figure 1F) was compared with that of the standard (Figure 1E) and has the same waveform.

The results are shown in Table 2; both FLOS and WEOS are rich in adenosine (2.12 and 283 mg/g) and polysaccharides (107.60 and 156.3 mg/g), while EEOS-50 and EEOS-95 have higher GABA, TPC, and TFC contents. It is worth noting that the contents of ergosterol, TPC, TFC, and GABA in EEOS-95 were higher than those in other extracts, with contents of 18.6, 2.28, 2.14, and 18.60 mg/g, respectively. Therefore, we hypothesized that EEOS-95 might be a potential extract of SCOS with anti-inflammatory effects. The main reason for the “non-detected” polysaccharides in EEOS-95 is that polysaccharides are not soluble in 95% alcohol, and therefore, the extract (EEOS-95) extracted with 95% ethanol did not contain polysaccharides.

### 3.2. Cytotoxicity Evaluation of SCOS

The cytotoxicity of SCOS mycelial extracts and FLOS (50–1000 µg/mL) in BV2 microglial cells was evaluated using MTT analysis after 24 h incubation. Table 3 indicates that the cell viability of all extracts was > 90% in BV2 cells at concentrations up to 1000 (µg/mL). Thus, there was no toxic effect when BV2 cells were treated with 50 to 1000 (µg/mL) of SCOS extracts. The results also indicated that when the concentration was <500 µg/mL, all SCOS samples increased the cell viability in a dose-dependent manner. In particular, at concentrations of 50 and 500 μg/mL EEOS-95, the percentage of cell viability enhanced from 116.60% to 196.50%, an increase of approximately 68.52%. At a concentration of 1000 µg/mL, the cell viability of all the samples began to show a decreasing trend.

### 3.3. Effect of O. sinensis Mycelia on Nitric Oxide

We examined the effect of various extracts of SCOS on LPS-induced NO production by Griess reagent assay (Table 4). BV2 cells were stimulated with LPS (1 μg/mL), and the NO level significantly increased (0.78 nmol/10^6^ cells) compared with the control (0.33 nmol/10^6^ cells) (*p* < 0.05). Our results also indicated that cells pre-treated with different concentrations (50–500 µg/mL) of extracts from SCOS mycelia—WEOS, EEOS-50, and EEOS-95—significantly suppressed the NO production by LPS-stimulated BV2 cells in a dose-dependent manner (*p* < 0.05); the inhibition effects at 500 µg/mL were 42, 44, and 71%, and EEOS-95 had the highest inhibition effect on NO production. FLOS also inhibited 42% of NO levels; however, treatment doses had no significant difference. (*p* < 0.05). Thus, we chose EEOS-95 to proceed with in the following anti-inflammation evaluation.

### 3.4. Anti-Inflammation Effect of EEOS-95 on LPS-Induced BV2 Microglial Cells

We further tested EEOS-95, which affected the cytokine production induced by LPS in BV2 microglial cells. The levels of IL-1β, TNF-α, IL-6, and PGE_2_ were determined by an ELISA kit. As shown in Table 5, LPS markedly increased IL-1β, IL-6, TNF-α, and PGE_2_ levels, as compared with untreated controls (*p* < 0.05); however, all of these pro-inflammatory cytokines were significantly decreased by EEOS-95 (50–500 µg/mL) in a concentration-dependent manner. In addition, at a concentration of 500 µg/mL, the inhibitory effects of EEOS-95 on IL-1β, IL-6, TNF-α, and PGE2 production (96.60, 70.33, 49.83, and 43.57%, respectively, *p* < 0.05) were significantly better than that of 1 mM MT (30.22, 61.60, 11.40, and 6.70%, respectively, *p* < 0.05). The inhibition ratio of EEOS-95 on different cytokines showed that EEOS-95 had the best inhibition effect on PGE2, followed by IL-1β, TNF-α, and IL-6, with inhibition rates of 90, 60, 50, and 40%, respectively, *p* < 0.05, at a concentration of 500 µg/mL (Figure 2).

### 3.5. Effect of EEOS-95 on the Expression of Inflammatory Proteins in LPS-Induced BV2 Microglial

Western blot analyses were performed to determine whether EEOS-95 had a direct effect on the pro-inflammatory factor in LPS-induced BV2 microglial. As shown in Figure 3A, LPS significantly induced the expression of inflammation-associated proteins such as iNOS, COX-2, and NF-κB, which were dose-dependently (50–500 µg/mL) reduced by EEOS-95 (*p* < 0.05). At a concentration of 500 µg/mL, EEOS-95 showed a 34, 55, and 20% decrease in protein expression of iNOS, COX-2, and NF-κB, respectively, compared to the LPS-treated group alone (Figure 3B–D, *p* < 0.05). Pretreatment of BV2 cells with 1 mM melatonin also reduced the expression levels of iNOS and COX-2 by 12% and did not affect NF-κB compared with the LPS group only.

### 3.6. Effect of EEOS-95 on the Expression of Anti-Inflammatory Protein in LPS-Induced BV2 Microglial Cells

As per the results shown in Figure 4, cells incubated with EEOS-95 significantly increased the expression levels of antioxidant factors (HO-1, Nrf-2, and PPAR-γ) in the LPS-induced BV2 microglial cells. The protein expression level of HO-1, Nrf-2, and PPAR-γ in BV2 cells treated with EEOS-95 at 25–250 μg/mL showed a significant increase (*p* < 0.05) compared with the control group and the LPS-treated group. In contrast, EEOS-95 at a concentration of 500 µg/mL increased the expression of the control group by more than 50% and was superior to that of the 1 mM MT-treated group.

## 4. Discussion

The wild caterpillar fungus *O. sinensis* is a traditional Chinese medicine, often found in Asia for its rich bioactive profile and diverse therapeutic applications, also has over 30 bioactive compounds that exhibit numerous beneficial effects, including anti-inflammatory, antioxidant, anti-tumor, immunomodulatory, and anti-osteoporotic activities, proving effective in tackling ailments like diabetes, liver cancer, and kidney cancer [15,25,26]. Overharvesting of wild *O. sinensis* has significantly reduced its annual production, causing a gap between rising demand and declining supply. This has fueled research into sustainable alternatives, including artificial cultivation using *C. militaris* and submerged culture techniques for natural *O. sinensis* [26,27,28,29,30]. These approaches focus on optimizing culture conditions, maximizing mycelial biomass production, and exploring polysaccharide extraction. While substantial research has investigated the pharmacological potential of SCOS, a vital knowledge gap remains concerning the detailed analysis of their extracted bioactive components. Our study employed aqueous and ethanolic solvents to extract bioactive compounds from SCOS to identify their main bioactive components and anti-inflammatory potential. We used melatonin (MT, N-acetyl-5-methoxytryptamine) for a positive control; MT is an animal hormone that exhibits physiological functions that can improve sleep and delay aging. Alvarez-García et al. found that 1 mM MT inhibited cytokine expression in malignant cells. Based on this evidence, we used a 1 mM concentration of MT on cells [31].

Yan et al. (2023) indicated that there is little difference in the nutritional components between a submerged culture and fruiting bodies [32]. The main biologically active ingredients isolated from the fruiting bodies of edible and medicinal fungi are polysaccharides, triterpenes, proteins, alkaloids, sterols, etc. These active compounds exhibit anti-tumor, immunoregulatory, antioxidant, hypoglycemic, and lipid-lowering effects [33,34]. During the mycelial growth of edible and medicinal fungi, the metabolism will secrete many nutrients and active ingredients (such as polysaccharides, GABA, ergosterol, flavonoids, vitamins, alkaloids, glycosides, and antibiotics) that can be extracted from both the mycelia and extracellular fluid (fermented liquid). They have anti-tumor, immunoregulatory, antioxidant, antibacterial, and antiviral activities.

Several bioactive compounds and their pharmacological properties have been studied in the mycelia and fermented liquid, such as polysaccharides, cordycepin, peptides, nucleosides, GABA, ergosterol, melanin, etc., which play an important role in immunomodulatory, antitumor, and antioxidant activities [35].

Our quantitative analysis of the dried powder of SCOS extract using RP-HPLC revealed six bioactive ingredients: adenosine, ergosterol, polysaccharide, TPC, TFC, and GABA. Briefly, adenosine belongs to nucleosides, which are essential bioactive compounds in organisms. The content of adenosine was 3.29 mg/g (EEOS-50), which was similar to previous reports (3.06 mg/g) and remarkably higher than that of natural *O. sinensis* [15]. Zhang et al. also pointed out that adenosine provides neuronal protection in NDs such as Alzheimer’s (AD) and Parkinson’s disease (PD), suggesting its value as a marker for quality control in an *O. sinensis* submerged fermentation [36]. However, these diverse bioactive compounds warrant further investigation into their potential health benefits and applications.

Ergosterol is an essential component of fungal cell membranes and is called the main fungal sterol; its function is similar to cholesterol in animals, which can be converted to vitamin D2 under ultraviolet radiation or sunlight. The vegetarian body needs be provided with vitamin D every day through a source of mushrooms; thus, ergosterol is interesting in further research [37]. The pharmacological effects of ergosterol have also been reported as antioxidant, anti-inflammatory, anti-neurodegenerative, antimicrobial, anticancer, antidiabetic, etc. [38,39,40,41,42,43,44,45,46]. As reported by Peng and colleagues, ergosterol is the active compound of cultured mycelium *C. sinensis*, which has inhibited liver fibrosis [45]. Herein, we confirm that ergosterol is only present in EEOS-95 (18.60 mg/g), which is six times higher than other published data in the literature (3.20 mg/g) [14]. Thus, the potential of phytol-ergosterol from the SCOS will be interesting for future biomedical applications.

Ergosterol exerts its anti-neuroinflammatory activity via the TLR4/NF-κB-dependent pathway. Therefore, exploring the potential for developing ergosterol into a novel drug for treating AD is viable [40]. Ergosterol also inhibits NF-κB luciferase activity in RAW246.7 macrophages [47]. Moreover, ergosterol binds directly to the active site of NF-κB p65 to restrain the phosphorylation and degradation of IκB-α and thus block the phosphorylation of NF-κB p65 [48]. Furthermore, ergosterol displays a significant anti-inflammatory effect on LPS-induced human monocytic cells through the inhibition of MyD88 (which is a central node of the inflammatory signaling pathway), VCAM-1 expression, and cytokine (IL-1β, IL-6, and TNF-α) production [49,50].

GABA is a non-protein amino acid that has been biosynthesized from glutamic acid in the human body. Boonstra’s study indicated that GABA is considered the major inhibitory neurotransmitter in the central nervous system, which has a lot of potential for various parts of the nervous system, including the cerebellum, hippocampus, hypothalamus, striatum, and spinal cord [51]. A previous study pointed out that biosynthetic GABA intake may have a beneficial effect on stress reduction and sleep improvement [52]. Surprisingly, the content of GABA in EEOS-95 (18.60 mg/g) is much higher than that found in *C. militaris* (0.0686–0.180 mg/g) and *C. sinensis* (0.220 mg/g) [27,53]. Thus, the GABA-rich SCOS will be a potential source of natural neuroprotection.

Evidence indicates that polysaccharides can promote cell proliferation, scavenge cellular ROS, and downregulate the secretion of pro-inflammatory cytokines [54,55]. In the cell viability test, we also found that FLOS, WEOS, EEOS-50, and EEOS-95 promoted cell proliferation (Table 3), presumably related to the polysaccharides in the samples. However, polysaccharides were not detected in EEOS-95, but its cell viability was higher than that of other extracts, suggesting that EEOS-95 may contain other cell proliferation components. Ergosterol is regarded as a “fungal hormone” that can stimulate growth and proliferation. Many studies have shown that ergosterol is essential for mitochondrial DNA maintenance in fungi, the same function cholesterol performs in humans [56,57,58]. Ergosterol also exerts a differential effect on Androgen-dependent LNCaP and Androgen-independent DU-145 cancer cells; ergosterol showed an antiproliferative effect on LNCaP and a proliferative effect on DU-145. Thus, the promoted cell proliferative effect of EEOS-95 in BV2 cells (Androgen-independent cells) should benefit from the ergosterol.

The TPC and TFC contents of fermented liquid (FLOS) and all mycelia extracts of the *O. sinensis* submerged culture were 1.57~2.28 mg Ga/g (TPC) and 1.17~2.14 mg Que/g (TFC). Tran et al. [59]. pointed out that the amounts of gallic acid, quercetin, quercitrin, and hesperidin dominated the *O. sobolifera* extracts at 193.60, 142.07, 544.53, and 110.08 µg/g, respectively, out of a total of a 990.27 µg/g dry weight of the active phenolic fraction, and these phenolic compounds of *O. sobolifera* extract were responsible for renal injury prevention [59]. Thus, we suggested that the TPC and TFC present in the SCOS extracts enhance the benefit of the pharmacological properties of SCOS for functional foods. Our results revealed that EEOS-95 was the most potent extract in SCOS by inhibiting the NO production induced by LPS in BV2 cells, which is related to their high contents of active compounds (ergosterol, TPC, TFC, and GABA).

Activation of the microglia leads to the production of excessive inflammatory molecules and deleterious consequences leading to neuronal death, which has been thought to contribute to the pathogenesis of NDs; the cause of acute injuries (stroke and traumatic brain injury); and chronic neurodegeneration (such as AD, PD, and chronic traumatic encephalopathy) [60]. Recent studies have reported that some compounds such as nucleoside, ergosterol, GABA, polysaccharides, and cordycepin isolated from *C. militaris* inhibited the production of NO, which reduced pro-inflammatory cytokines; they also possessed an effect of neuroprotection by inhibiting microglia-mediated inflammation in LPS-induced microglia BV2 cells [40,61]. In the present study, our results demonstrated that EEOS-95 significantly inhibited the levels of cytokines (IL-1β, IL-6, TNF-α, and PGE2) secreted and decreased the production of NO in LPS-stimulated BV2 cells. Therefore, using the natural ingredients obtained from EEOS-95 as therapeutics for neurodegenerative disorders with neuro-inflammation is possible.

Peroxisome proliferator-activated receptor gamma (PPAR-γ) is a subunit of the PPAR and is a ligand-activated nuclear transcription factor; if the brain is injured, PPAR-γ can be the “key” to cytoprotective stress responses and enhance the chances of cellular survival [60]. Furthermore, several recent studies have shown that PPAR-γ is also capable of inhibiting other transcription factors and has been implicated in the downregulation of the transcription and expression of related genes involved in proinflammatory cytokines (IL-1β, IL-6, and TNF-α), and neuro-inflammatory genes (COX-2 and iNOS), additionally, as it can inhibit active-NF-κB signals, playing a neuroprotective role for microglia [60,62].

Luteoloside (a flavonoid compound) significantly upregulated PPAR-γ and Nrf-2 and decreased the release of proinflammatory cytokines in focal cerebral ischemia in middle cerebral artery occlusion (MCAO) rats by inhibiting the NF-κB pathway [63]. Ergosterol isolated from *mushrooms* (*A. polytricha* and *C. militaris*) attenuates bisphenol A or LPS-induced BV2 microglial cell inflammation [42,43]. Zheng et al. indicated that a GABA-enriched *Moringa oleifera* leaf (MLFB) fermentation broth could also effectively alleviate the LPS-induced inflammatory response by inhibiting the secretions of pro-inflammatory cytokines and the anti-inflammatory activity might be related to the relatively high contents of GABA, flavonoids, phenolics, and organic acids in MLFB; its mechanism might be associated with the inhibition of TLR-4/NF-κB inflammatory signaling pathway activation [64]. Actually, treatment with EEOS-95 significantly inhibited the expression of NF-κB, iNOS, and COX-2 and enhanced the upregulation of Nrf-2 and HO-1; likewise, it can inhibit active-NF-κB signaling as well as play a neuroprotective role for microglia in the PPAR-γ/Nrf-2/HO-1/NF-κB signaling pathway, and ergosterol, GABA, flavonoids, and polyphenols may be responsible for this activity.

Additionally, the nuclear factor Nrf-2 is known as another transcription factor and a master regulator of detoxification and antioxidant regulation, which may play a main role in neuroprotective function [65]. Some documents suggested that it correlates with PPAR-γ and Nrf-2, which are exerted against oxidative stress, effectively reducing the inflammatory response by inhibiting NF-κB signals [66,67]. In addition, Duan et al. pointed out the synergistic effect of the PPAR-γ and Nrf-2 pathway to upregulate the expression of related genes and inhibit ferroptosis-induced neuronal injury in intracerebral hemorrhage rats in vitro and in vivo [67]. Furthermore, pro-oxidant HO-1 expression is upregulated by oxidative stress, nitric oxide, CO, and hypoxia. Choi and colleagues also highlighted that HO-1 is present and has a role in neurovascular diseases, such as age-related macular degeneration (AMD), ischemia-reperfusion injury, traumatic brain injury, and AD [68]. Abnormal HO-1 levels with Nrf-2 dysfunction are implicated in pathogenesis in neurovascular systems related to ischemia, trauma, and aging; thus, the Nrf-2/HO-1 signal mechanism is involved in development, oxidative stress responses, and anti-inflammation [69].

The antioxidative function of PPARγ was reported to be mediated by the transcriptional activation of several antioxidant genes such as HO-1, CAT, and manganese superoxide dismutase (MnSOD) through its direct association with the PPAR response elements of their promoter regions [70,71]. PPARγ was indeed able to suppress inflammation by transcriptional repression of many proinflammatory transcription factors and enzymes such as nuclear factor kappa B (NF-κB), signal transducer and activator of trancription-6 (STAT-6), activator protein 1 (AP-1), and cyclooxygenase-2 (COX-2), and induced nitric oxide synthase (iNOS) [72]. Thus, PPARγ has been regarded as a new anti-inflammatory and antioxidative pharmacotherapy target in many diseases adversely affected by oxidative stress and inflammation [73]. Therefore, we speculate that EEOS-95 has an effect on PPAR- γ, an activator that inhibits oxidative stress, and against neuronal inflammation through the synergistic actions of the expression of PPAR-γ, Nrf-2, and HO-1 pathways in LPS-stimulated microglia BV2 cells (Figure 5). Furthermore, EEOS-95 has exhibited strong pharmacological properties and exerted a potential anti-neuroinflammatory effect.

## 5. Conclusions

The findings demonstrate the potential of EEOS-95 to be developed into a functional food due to its anti-inflammatory effects. EEOS-95 not only upregulated PPAR-γ/Nrf-2/HO-1 and downregulated NF-κB/COX-2/iNOS pathways, but it also decreased pro-inflammatory cytokines (IL-1β, IL-6, and TNF-α). We also found six types of bioactive components involving adenosine, ergosterol, polysaccharides, GABA, TPC, and TFC present in the extracts of SCOS. The synergism of these phytochemicals would contribute to the pharmacological properties in LPS-induced BV2 microglia cells. Thus, SCOS could be a potential source for neuroprotection. We propose that SCOS could be used as raw material for functional products or nutraceuticals.

## Figures and Tables

**Figure 1 jof-10-00523-f001:**
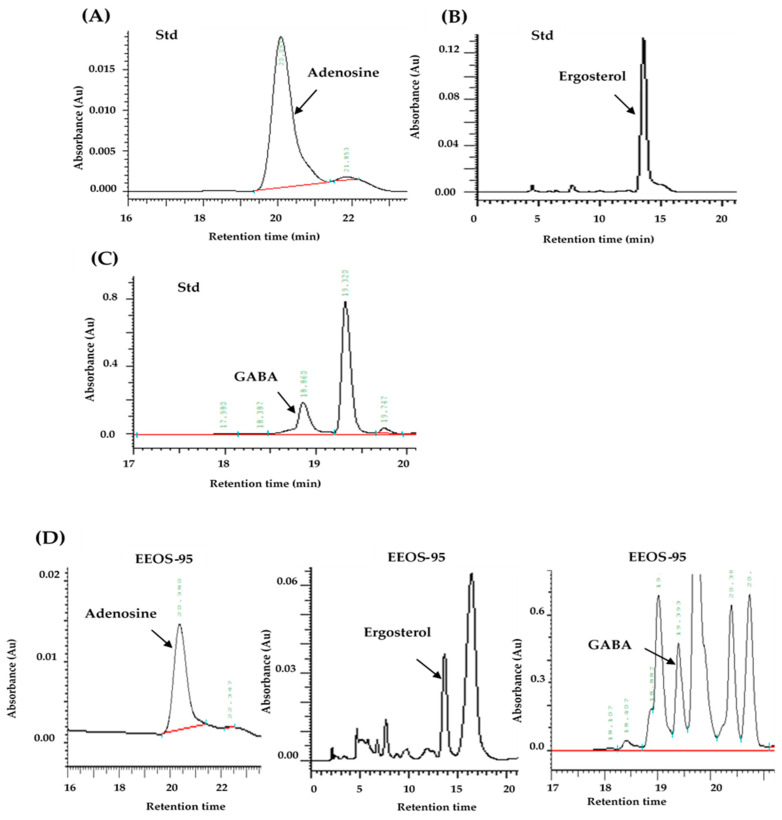
HPLC chromatograms and full-wavelength spectrum of adenosine, ergosterol, GABA, and EEOS-95. The HPLC chromatograms of standard adenosine (**A**), ergosterol (**B**), GABA (**C**) and EEOS-95 (**D**) were detected. A diode array detector (DAD) was used to detect the full-wavelength (200–400 nm), comparison standard waveforms (adenosine, ergosterol, and GABA) (**E**), and EEOS-95 (**F**). EEOS-95: 95% ethanol extract of SCOS.

**Figure 2 jof-10-00523-f002:**
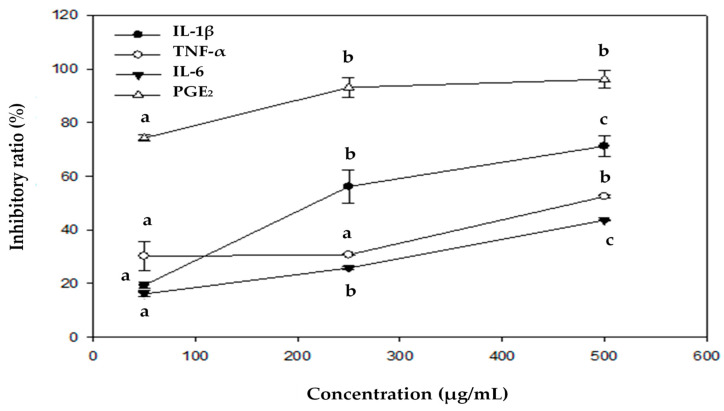
The inhibitory ratio of EEOS-95 on LPS-induced cytokine production in BV2 microglial cells. Values (means ± SD, *n* = 3 for the test groups) not sharing the same superscript letter are significantly different (*p* < 0.05). Interleukins-1β (IL-1β), interleukins-6 (IL-6), tumor necrosis factor-α (TNF-α), and prostaglandin E2 (PGE2).

**Figure 3 jof-10-00523-f003:**
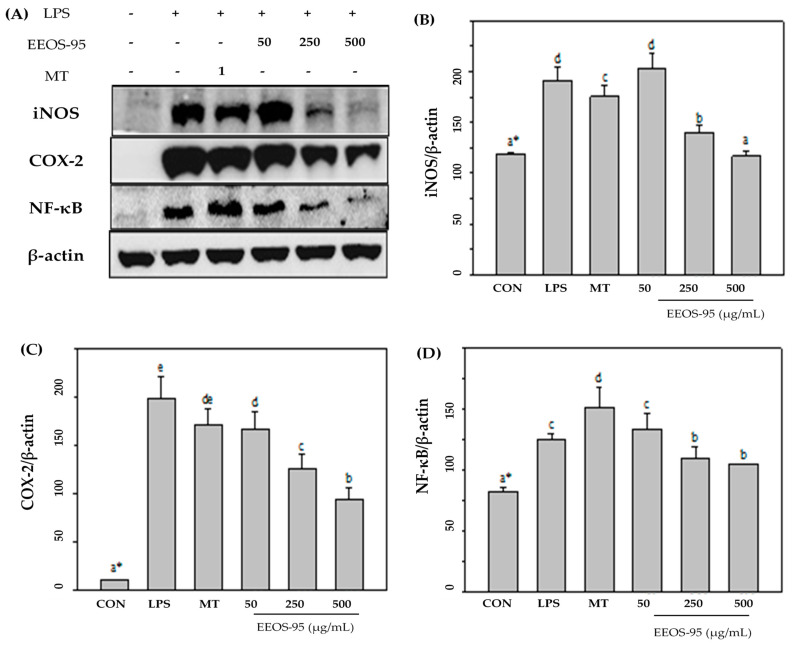
Effect of EEOS-95 on the expression levels of pro-inflammation factors in LPS-induced BV2 microglial cells. Cells were treated with EEOS95 (50–500 μg/mL) and MT (1 mM Melatonin) for 24 h and incubated with 1 µg/mL LPS for 24 h. (**A**) Protein expression of inducible nitric oxide synthase (iNOS), cyclooxygenase-2 (COX-2), nuclear factor-κB (NF-ĸB), and β-actin were detected in the cytoplasm by a Western blotting assay. The quantitative values of (**B**) iNOS, (**C**) COX-2, and (**D**) NF-ĸB were analyzed using ImageJ software v1.8.0. * Values (means ± SD, *n* = 3 for the test groups) not sharing the same superscript letter are significantly different (*p* < 0.05).

**Figure 4 jof-10-00523-f004:**
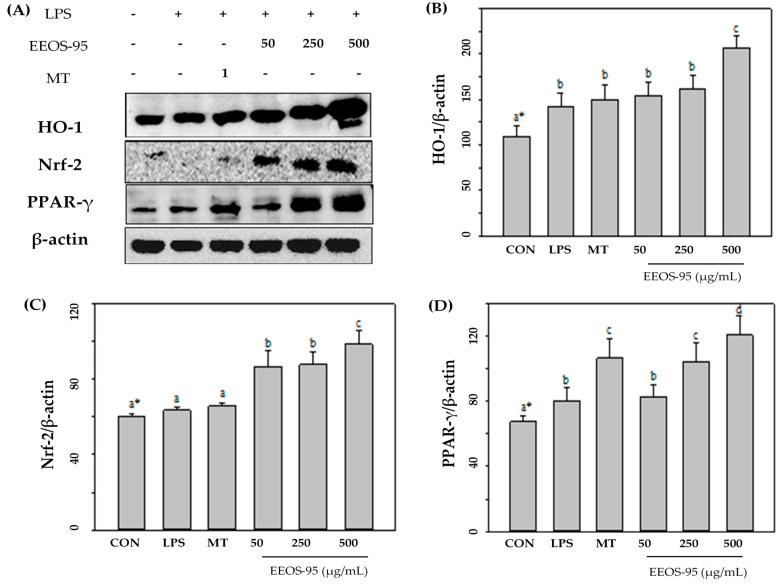
Effect of EEOS-95 on expression levels of antioxidant factors in LPS-induced BV2 microglial cells. Cells were treated with EEOS95 (50–500 μg/mL) and MT (Melatonin 1 mM) for 24 h and incubated with LPS 1 µg/mL for 24 h. (**A**) Protein expression of heme oxygenase-1 (HO-1), nuclear-related factor 2 (Nrf-2), peroxisome proliferator-activated receptors-γ (PPAR-γ), and β-actin were detected in the cytoplasm by a Western blotting assay. The quantitative values of (**B**) HO-1, (**C**) Nrf-2, and (**D**) PPAR-γ were analyzed using ImageJ software v1.8.0. * Values (means ± SD, *n* = 3 for the test groups) not sharing the same superscript letter are significantly different (*p* < 0.05).

**Figure 5 jof-10-00523-f005:**
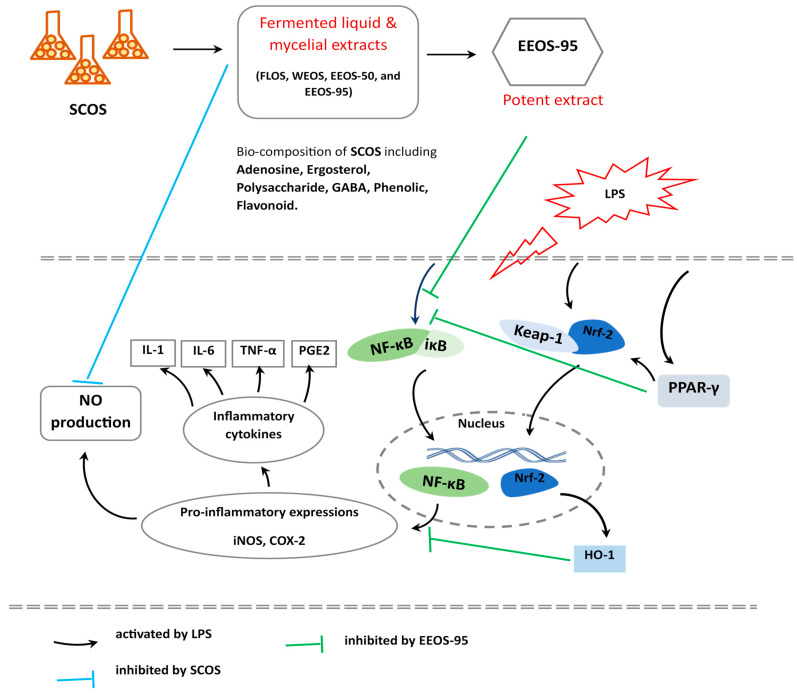
Schematic diagram of the mechanism of speculated anti-inflammatory ability of EEOS-95. FLOS: fermented liquid of SCOS, WEOS: hot water extract of SCOS, EEOS-50: 50% ethanol extract of SCOS, and EEOS-95: 95% ethanol extract of SCOS, mycelial submerged culture of *O. sinensis* H101 (SCOS). Nitric oxide (NO), interleukins-1β (IL-1β), interleukins-6 (IL-6), tumor necrosis factor-α (TNF-α), prostaglandin E2 (PGE2)), inducible NO synthase (iNOS), cyclooxygenase-2 (COX-2), nuclear factor kappa B (NF-κB)), nuclear factor-erythroid 2-related factor 2 (Nrf-2), peroxisome proliferator-activated receptors (PPAR-γ), and heme oxygenase-1 (HO-1). γ-aminobutyric acid (GABA), and lipopolysaccharide (LPS).

**Table 1 jof-10-00523-t001:** Correlation coefficients (R^2^), retention time (R.T.), detection limits (LD), linear calibration curve, relative standard deviation (RSD), number of data points, and detection methods for the standard chemicals. Values are means ± SD, *n* = 3 for the test groups, *p* < 0.05.

Standard Chemicals	R^2^	R.T. (min)	LD	Linear Equation	RSD %	Number of Data Points	Detection Methods
Adenosine	0.9971	20.31	40 μg/mL	y=12255.09x−3756.04	100.6006	5	RP-HPLC
Ergosterol	0.9148	14.03	6.0 μg/mL	y=753758x−1185200	68.3628	5	RP-HPLC
Polysaccharide	0.9975	-	0.10 mg/mL	y=19.8894x+0.1320	67.7326	5	Colorimetric
Total Polyphenols (mg Ga/g)	0.9952	-	0.10 mg/mL	y=2.5844x+0.0407	68.0186	5	Colorimetric
Total Flavonoids (mg Que/g)	0.9973	-	10 μg/mL	y=0.00237x+0.0514	34.9659	5	Colorimetric
GABA	0.9441	19.32	200 μg/mL	y=17039x+430827	57.3752	5	RP-HPLC

**Table 2 jof-10-00523-t002:** Active compounds of various extracts from SCOS.

Compounds (mg/g)	SCOS Extracts ^1^			
	FLOS	WEOS	EEOS-50	EEOS-95
Adenosine	2.12 ± 0.09 ^b,2^	2.83 ± 0.05 ^c^	3.29 ± 0.29 ^d^	0.19 ± 0.01 ^a^
Ergosterol	ND	ND	ND	18.60 ± 0.70
Polysaccharide	107.60 ± 10.20 ^b^	156.30 ± 7.10 ^c^	28.50 ± 7.70 ^a^	ND
Total Polyphenols (mg Ga/g)	1.57 ± 0.09 ^a^	1.71 ± 0.07 ^bc^	1.77 ± 0.15 ^c^	2.28 ± 0.05 ^d^
Total Flavonoids (mg Que/g)	1.34 ± 0.01 ^b^	1.17 ± 0.02 ^a^	1.65 ± 0.01 ^c^	2.14 ± 0.06 ^d^
GABA	3.70 ± 0.30 ^a^	12.60 ± 0.80 ^b^	13.20 ± 0.60 ^b^	18.60 ± 0.50 ^c^

^1^ FLOS: fermented liquid of SCOS, WEOS: hot water extract of SCOS, EEOS-50: 50% ethanol extract of SCOS, and EEOS-95: 95% ethanol extract of SCOS. ND: non-detected. ^2^ Values (means ± SD, *n* = 3 for the test groups) not sharing the same superscript letter in a row are significantly different (*p* < 0.05).

**Table 3 jof-10-00523-t003:** The cytotoxicity of various extracts of SCOS on BV2 cells.

Concentration (µg/mL) ^1^	Cell Viability (% of CON)			
	FLOS ^2^	WEOS	EEOS-50	EEOS-95
50	89.80 ± 13.20 ^a,3^	95.90 ± 7.20 ^a^	94.30 ± 11.90 ^a^	116.60 ± 8.60 ^a^
100	106.40 ± 25.60 ^a^	96.70 ± 1.40 ^a^	102.20 ± 6.00 ^a^	137.30 ± 12.60 ^b^
250	119.40 ± 18.30 ^ab^	108.80 ± 5.10 ^a^	116.80 ± 6.20 ^ab^	171.70 ± 12.90 ^cd^
500	134.20 ± 11.30 ^b^	123.70 ± 11.90 ^b^	125.70 ± 4.40 ^b^	196.50 ± 13.80 ^d^
1000	114.00 ± 23.40 ^ab^	116.50 ± 8.40 ^ab^	123.00 ± 1.00 ^b^	175.90 ± 3.00 ^c^

^1^ Cells were pretreated with various extracts of SCOS (50–1000 µg/mL) for 24 h. Viability was measured by MTT assay. Control (CON) was treated with the culture medium of mycelia. ^2^ FLOS: fermented liquid of SCOS, WEOS: hot water extract of SCOS, EEOS-50: 50% ethanol extract of SCOS, and EEOS-95: 95% ethanol extract of SCOS. ^3^ Values (means ± SD, *n* = 3 for the test groups) not sharing the same superscript letter in a column are significantly different (*p* < 0.05).

**Table 4 jof-10-00523-t004:** Effect of various extracts of SCOS on LPS-induced nitrite production in BV2 microglial cells.

Treatments ^1^		Nitrite Concentration (nmol/10^6^ Cells)			
			50	250	500
CON		0.33 ± 0.05 ^a,4^	-	-	-
LPS (1 µg/mL)		0.78 ± 0.12 ^d^	-	-	-
MT (1 mM) ^2^	+LPS	0.45 ± 0.03 ^b^	-	-	-
FLOS ^3^		-	0.55 ± 0.00 ^c^	0.59 ± 0.07 ^c^	0.59 ± 0.07 ^c^
WEOS		-	0.75 ± 0.02 ^d^	0.74 ± 0.02 ^d^	0.59 ± 0.05 ^c^
EEOS-50		-	0.76 ± 0.11 ^d^	0.66 ± 0.05 ^cd^	0.58 ± 0.10 ^c^
EEOS-95		-	0.64 ± 0.16 ^cd^	0.53 ± 0.09 ^c^	0.46 ± 0.02 ^b^

^1^ Cells were pretreated with 1 mM melatonin (MT) or various extracts of SCOS (µg/mL) for 24 h and then incubated with LPS 1 µg/mL for 24 h. Control (CON) was treated with the culture medium of mycelia. ^2^ Melatonin (MT, N-acetyl-5-methoxytryptamine) is an animal hormone that exhibits physiological functions such as improving sleep, delaying aging, etc. MT is a positive control. ^3^ FLOS: fermented liquid of SCOS, WEOS: hot water extract of SCOS, EEOS-50: 50% ethanol extract of SCOS, and EEOS-95: 95% ethanol extract of SCOS. ^4^ Values (means ± SD, *n* = 3 for the test groups) not sharing the same superscript letter in a column are significantly different (*p* < 0.05).

**Table 5 jof-10-00523-t005:** Effect of EEOS-95 on LPS-induced cytokines in BV2 microglial cells.

Treatments ^1^		Cytokines			
		IL-1β (pg/mL)	TNF-α (ng/mL)	IL-6 (ng/mL)	PGE_2_ (pg/mL)
CON		4.29 ± 1.30 ^a,3^	13.71 ± 0.69 ^a^	0.00 ± 0.00 ^a^	97.60 ± 5.50 ^a^
LPS (1 µg/mL)		76.66 ± 3.47 ^f^	70.54 ± 4.89 ^d^	608.22 ± 17.40 ^e^	933.90 ± 12.30 ^e^
MT (1 mM) ^2^	+ LPS	54.79 ± 0.94 ^d^	35.53 ± 3.58 ^b^	538.87 ± 28.22 ^de^	877.90 ± 10.30 ^d^
50		62.86 ± 3.90 ^e^	49.58 ± 0.75 ^c^	510.21 ± 7.09 ^cd^	308.20 ± 9.80 ^c^
250		36.59 ± 1.54 ^c^	53.10 ± 3.53 ^c^	451.58 ± 2.73 ^c^	155.40 ± 30.40 ^b^
500		25.76 ± 0.61 ^b^	42.22 ± 3.22 ^bc^	343.20 ± 1.86 ^b^	126.00 ± 27.90 ^a^

^1^ Cells were pretreated with 1 mM melatonin or 50–500 (µg/mL) of EEOS-95: 95% ethanol extracts of SCOS for 24 h and then incubated with LPS 1 µg/mL for 24 h. ^2^ Melatonin (MT) is a positive control; the control (CON) was treated with the culture medium of mycelia. ^3^ Values (means ± SD, *n* = 3 for the test groups) not sharing the same superscript letter in a column are significantly different (*p* < 0.05).

## Data Availability

The data presented in this study are available on request from the corresponding author.

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
