# Peer review of "The Anti-Inflammatory Effects and Mechanism of the Submerged Culture of Ophiocordyceps sinensis and Its Possible Active Compounds"

_jof, 2024, doi:10.3390/jof10080523_

Round 1
Reviewer 1 Report (Previous Reviewer 2)
In this study the authors evaluated the pharmacological effects of the fruiting body of Ophiocordyceps sinensis (Os) against inflammation in LPS-stimulated microglial BV2 cells. The authors studied the effects of fermented liquid (FLOS), hot water (WEOS), and 50-95% (EEOS-15 50, EEOS-95) ethanol extracts of OMSC and their anti-inflammatory effects and potential mechanisms in LPS-stimulated microglial BV2 cells. The resubmitted work is revised well based on the previous comments. This research work has potential and beneficial for the researchers of interest.
Comments
Manuscript ID: jof-3046912
Type: Article
Title: The anti-inflammatory effects and mechanism of the mycelial submerged culture of Ophiocordyceps sinensis and its possible active compounds.
Comments
In this study the authors evaluated the pharmacological effects of the fruiting body of Ophiocordyceps sinensis (Os) against inflammation in LPS-stimulated microglial BV2 cells. The authors studied the effects of fermented liquid (FLOS), hot water (WEOS), and 50-95% (EEOS-15 50, EEOS-95) ethanol extracts of OMSC and their anti-inflammatory effects and potential mechanisms in LPS-stimulated microglial BV2 cells.
The resubmitted work is revised well based on the previous comments. This research work has potential and beneficial for the researchers of interest. However, I have few concerns before a decision can be made by the respective editors for its possible acceptance and publication.
Minor comments
Title:
The title can be modified as “The anti-neuroinflammatory effects of the mycelial submerged culture of Ophiocordyceps sinensis and its active compounds on lipopolysaccharide-stimulated BV-2 microglia”
Abstract:
-Page 1, line 25-26: “Therefore, we hypothesized that EEOS-95 can prevent neurodegenerative diseases, but the effect on the application in vivo should be further investigated”. The authors should revise the statement. The authors should restrict the use of “neurodegeneration” as no in vivo work is done and no direct evidence. Therefore, the authors should only emphasize on the present work done. Instead they can mention “neuroinflammation-mediated neurodegenerative disorders”.
Figures
-Figure2. In legends, all the abbreviations used (e.g.: IL-1B, TNF-a, IL-6 and PGE2) should be given in full form.
-Figure5: In legends, all the abbreviations used in the diagram should be given in full form.
Discussion
-Page 12, line 367, Alzheimer’s and Parkinson’s disease: write full forms at first appearance and then later abbreviate them. Check throughout the article.
Author Response
Dear reviewer,
Thank you very much for taking the time to review this manuscript. Please find the detailed responses below and the corrections highlighted in red in the re-submitted MS files.
Sincerely yours,
Tuzz-Ying Song

Reviewer 2 Report (New Reviewer)
1. The authors write that "the anti-inflammatory effects and active components of Os mycelial submerged culture (OMSC) are not known." This is not correct, such works are known and they should be given in the Introduction. Here are some similar works:
a. Chau, H. B., Hoang, N. T., My, N. T. T., Duy, B. L., Nam, T. V. H., Hang, H. T. B., & Hiep, D. M. (2018). Effect of rice bran oil on mycelial biomass production, biosynthesis and bioactivities of polysaccharides by Ophiocordyceps sinensis fungus. Vietnam Journal of Science and Technology, 56(4A), 74-82.
b. Kuo, M. C., Chang, C. Y., Cheng, T. L., & Wu, M. J. (2007). Immunomodulatory effect of exo-polysaccharides from submerged cultured Cordyceps sinensis: enhancement of cytokine synthesis, CD11b expression, and phagocytosis. Applied microbiology and biotechnology, 75, 769-775.
c. Liu, Z., Li, P., Zhao, D., Tang, H., & Guo, J. (2011). Anti-inflammation effects of Cordyceps sinensis mycelium in focal cerebral ischemic injury rats. Inflammation, 34, 639-644.
d. Huang, D., Meran, S., Nie, S. P., Midgley, A., Wang, J., Cui, S. W., ... & Phillips, A. O. (2018). Cordyceps sinensis: anti-fibrotic and inflammatory effects of a cultured polysaccharide extract. Bioactive carbohydrates and dietary fibre, 14, 2-8.
e. Zhang, Y., Li, K., Zhang, C., Liao, H., & Li, R. (2023). Research Progress of Cordyceps sinensis and Its Fermented Mycelium Products on Ameliorating Renal Fibrosis by Reducing Epithelial-to-Mesenchymal Transition. Journal of Inflammation Research, 2817-2830.
2. It is necessary to adhere to commonly used terms. The use of incorrect terms leads to inaccuracies in the preparation of the manuscript and to difficulties in reading it.
a. What is "mycelial submerged culture"? Is it a submerged mycelium or culture liquid? Or submerged culture of fungus?
b. A submerged culture of fungus is a its submerged mycelium and culture liquid. These terms should be used when describing research methods. It is better to exclude the term "fermented liquid".
c. The authors write that "the OMSC extracts were lyophilized to obtain fermented liquid of Os (FLOS)..." It is unclear how lyophilization of extracts leads to the production of dry culture liquid.
3. It is unclear how the results shown in Table 4 allowed the authors to conclude that the inhibitory effects were 25, 26 and 41%.
4. The Discussion section needs to be revised, because now it looks more like an Introduction.
1. In JoF, it is advisable to use the Latin name of the species.
2. In the section "Materials and methods", it should be indicated by which method the submerged biomass was separated from the culture liquid, and the raw or dried mycelium was extracted. In addition, it is desirable to provide information about the morphology of the submerged mycelium.
3. Remarks on Table 3.
a. The table name "Effect of various extracts of OMSC on cytotoxicity..." is unfortunate. The cytotoxicity of the preparations obtained was studied.
b. It is necessary to indicate what was used as a control, in relation to which cytotoxicity was evaluated.
4. It is necessary to give an explanation of the abbreviations LPS and GABBA at the first mention.
5. Please make sure that the genus name is abbreviated when repeated, and the species names are given with a small letter.
6. Please correct the typo on L. 263.
Author Response
Dear reviewer,
Thank you very much for taking the time to review this manuscript. Please find the detailed responses below and the corrections highlighted in red in the re-submitted MS files.
Sincerely yours,
Tuzz-Ying Song
Reviewer 3 Report (New Reviewer)
Ophiocordyceps sinensis has important medicinal value. The manuscript authored by Huang reported that the active ingredients in OMSC fermentation broth (FLOS), hot water (WEOS), and 50-95% (EEOS-50, EEOS-95) ethanol extracts, as well as their anti-inflammatory effects and potential mechanisms on LPS stimulated microglial BV2 cells. The results showed that EEOS-95 not only up-regulated the PPAR-γ/Nrf-2/HO-1 pathway, down-regulated the NF-κb/COX-2/iNOS pathway, but also reduced pro-inflammatory factors (IL-1β, IL-6, TNF-α). In addition, six bioactive components were discovered, including adenosine, ergosterol, polysaccharides, GABA, TPC, and TFC. However, the following issues need to be addressed.
1. Insufficient introduction of research background and insufficient clarification of the importance of Ophiocordyceps sinensis in anti-inflammatory research. The authors should increase the review of relevant literature, especially the research progress in recent years, highlighting the shortcomings of current research.
2. It is necessary to explain in R1 the reasons for the selected detection components and the role these components play in alleviating neuroinflammation
3. Further discussion is needed on the differences in biological active ingredients and effects between Cordyceps sinensis deep fermentation and fruiting bodies
4. In Fig. 5, according to the schematic diagram of the effects of OMSC and EEOS-95 on lipopolysaccharide stimulated 483 related BV2 microglia, the relationship between EEOS-95 and PPAR-γ/Nrf-2/HO-1 pathway, NF-κb/COX-2/iNOS pathway, and whether EEOS-95 upregulates Nrf-2/HO-1 directly or indirectly through PPAR-γ is not clearly explained. Similarly, it is not explained in the NF-κb/COX-2/iNOS pathway.
5. Lack of analysis on the differences in principal components caused by EEOS-95 and other extraction methods, as well as the role of differential components.
6. In result 3.1, why is it assumed that EEOS-95 is an extract of OMSCs with anti-inflammatory effects, rather than FLOS and WEOS with higher concentrations of polysaccharides? Polysaccharides have strong anti-inflammatory effects and there are many related studies. It is recommended to provide additional explanations.
7. In result 3.1, according to the data in Table 2 mentioned, the polysaccharide content of EEOS-95 is ND. Is it because in Method 2.2, the extract was centrifuged after soaking in 95% ethanol at room temperature for 24 hours, and the supernatant was concentrated under reduced pressure using a rotary evaporator? At this point, the polysaccharide alcohol precipitate was discarded. If the polysaccharide is not discarded, is it more effective in anti-inflammatory effects on cells? It is recommended to provide additional explanation.
8. In line 393, it is stated that FLOS, WEOS, and EEOS-50 have a promoting effect on cell proliferation, which may be related to the polysaccharides contained in the sample. The author should remove this part of the polysaccharides before conducting further analysis.
1. The experimental design and method description in the Materials and Methods section are not detailed enough, especially the extraction method, processing conditions, and experimental steps. The article mentions the use of ether extract EEOS95, but lacks detailed extraction steps and key parameters (such as extraction time, temperature, solvent ratio, etc.).
2. The data presented in the table is inconsistent. For example, the correlation coefficient, retention time, and detection limit of certain compounds are given, while the retention time of other compounds is not, resulting in incomplete data representation.
3. In terms of experimental design, the article did not clearly explain the selection criteria for specific extracts (FLOS, WEOS, EEOS-50, EEOS-95) used, nor did it prove the rationality of the concentration and treatment duration selected in the experiment.
4.The data in the chart lacks detailed explanation and statistical analysis, and the results section lacks sufficient statistical analysis results such as p-values and confidence intervals to support the significance and credibility of the experimental data.
5. In lines 139-140, the conditions for gradient elution of the method are not fully written. It is recommended to supplement them.
6. In terms of biological activity testing, the control group setting may not be comprehensive enough, lacking a positive control group or an appropriate control group (known to have anti-inflammatory effects) to compare the anti-inflammatory effects of EEOS95.
7. Repeated experiments and validation experiments were not mentioned in the anti-inflammatory analysis experiment.
8. In result 3.3, why was MT chosen as the positive control group for reducing lipopolysaccharide induced BV2 microglial inflammation, and why was a concentration of 1mM chosen? Please explain.
9. In result 3.4, why only pro-inflammatory factors were studied, without including anti-inflammatory factors? Please provide additional experiments or explain.
Author Response
Dear reviewer,
Thank you very much for taking the time to review this manuscript. Please find the detailed responses below and the corrections highlighted in red in the re-submitted MS files.
Sincerely yours,
Tuzz-Ying Song

Reviewer 4 Report (New Reviewer)
In the manuscript:" The anti-inflammatory effects and mechanism of the mycelial submerged culture of Ophiocordyceps sinensis and its possible active compounds", the authors presented the results of their research concerning the anti-inflammatory potential and bioactive components contained in selected extracts derived from Ophiocordyceps sinensis submerged grown mycelium, with special emphasis to EEOS-95, which emerged as the most potent. Given that natural resources are limited, and the demand for this mushroom is increasing, this study is significant because it contributes to the importance of artificial cultivation of mycelium, which has proven to be very rich in bioactive substances.
Based on the obtained results, the authors also suggest the potential application of the most active anti-inflammatory extract as a component of functional food, but the realization of this possibility would require more extensive research.
I want to emphasize that I made grammatical changes/changes in sentence constructions in places where I considered it necessary to keep the meaning that the authors wanted.
All my suggestions are attached.

Author Response
Dear reviewer,
Thank you very much for taking the time to review this manuscript. Please find the detailed responses below and the corrections highlighted in red in the re-submitted MS files.
Sincerely yours,
Tuzz-Ying Song

Round 2
Reviewer 2 Report (New Reviewer)
I thank the authors for their attentive attitude to my remarks. I have no serious comments on the revised text.
Minor remarks
1. I suggest that the authors replace "Sen researchers" (L 42-43) with "Sen et al." or "Sen and co-authors".
2. I think that Figure 5 should be moved from the "Conclusion" section and the "Discussion" section

Author Response
Dear reviewer,
We thank for reviewer comments and suggests. Please find the detailed responses below and the corrections highlighted in green in the re-submitted MS files.
Sincerely yours,
Tuzz-Ying Song, PhD., Professor

This manuscript is a resubmission of an earlier submission. The following is a list of the peer review reports and author responses from that submission.
Round 1
Reviewer 1 Report
The authors have determined the bioactive compounds of extracts from Ophiocordyceps sinensis submerged culture, and the possible anti-inflammatory mechanism of EEOS-95 in the LPS‑induced BV2 microglial.
The article is well writtern and English language is quilt good. But I have some questions and suggestions.
1. From line 54 to 61, there is too much discription of Neurodegenerative diseases.
2. In table 4, what is “CON” represent for? It is better to explain in the table note.
3. In the "discussion", whether it is need to discuss each bioactive compound here?
Author Response
Dear Reviewer,
Thank you very much for taking the time to review this manuscript. Please see the detailed response below and the corresponding corrections highlighted in the resubmitted files.
Sincerely yours,
Tuzz-Ying Song, PhD., Professor

Reviewer 2 Report
Comments
Type of manuscript: Article.
Title: The preventive potential of the mycelial submerged culture of
Ophiocordyceps sinensis against neurodegenerative diseases and its possible
active compounds
Journal: Journal of Fungi
Manuscript ID: jof-2988056.
In this research, the authors evaluated the preventive effects of mycelial submerged culture of Ophiocordyceps sinensis against neuroinflammation. The authors studied the effects of fermented liquid (FLOS), hot water (WEOS), and 50-95% (EEOS-15 50, EEOS-95) ethanol extracts of OMSC and their anti-inflammatory effects and potential mechanisms in LPS-stimulated microglial BV2 cells. This research work has potential and is beneficial for the researchers of interest in this area.
However, I have few concerns before a decision can be made by the respective editors for its possible acceptance and publication.
English Language:
Revisions in English grammar and tense corrections have to be made. I recommend the authors proofread the final draft with an English-speaking scientific colleague or using English-speaking Scientific editing service to reduce grammar and tense errors.
Comments
Title: In the title, the authors mention “The preventive potential of the mycelial submerged culture of Ophiocordyceps sinensis against neurodegenerative diseases and its possible active compounds”, but the focus of the work is mainly anti-inflammation or anti-neuroinflammation mechanisms. Therefore, the title should be modified accordingly in relation to the work performed. There is no in vivo work performed for direct evidence in neuroprotection or neurodegeneration
Abstract:
-Line 25, the authors mentioned that “we hypothesized that EEOS-95 can prevent neurodegenerative diseases, but the effect on the application in vivo should be further investigated”. Based on this, the title does not fit for the present study. “Change of title as per the work done is necessary.
Introduction:
-the statements in Lines 33; 34-35, 35-36, 36-37, need solid references in support of the claims.
- Lines 37-39; reframe the statements for better understanding.
- Lines 88-89; reframe the statements for better understanding.
Materials and Methods
-section 2.3.1; line 123-124; correction the sentence formation “Adenosine was determined by applying the method described in Chang et al., with a slightly modified”??. It should be “with slight modification”.
-line 193; write full forms at first appearance and then later abbreviate them “iNOS, COX‑2, NF‑κB, HO‑1, Nrf‑2, PARP‑γ,”. Also check throughout manuscript.
Discussion:
-Line 473-474; Is the statement correct “Furthermore, EEOS-95 has exhibited strong pharmacological properties and exerted a potential neuro-inflammatory effect; -- Is it neuroinflammatory or anti-neuroinflammatory????. Please check.
-Line 474-477, “so, we will continue to investigate deeply explore the molecular mechanism by which PPAR-γ/Nrf-2/HO-1 pathway to neuro-protection in vivo, safe-dose, as well as preclinical and clinical in future studies of 476 95% ethanol extract of OMSC mycelia.“. Future work Statements Not necessary in a research article.
Focus on the present results and conclude.
Conclusions:
-Line 479-482; “The findings demonstrate the potential of EEOS-95 to be developed into functional food due to its against neuro-inflammatory by upregulated PPAR-γ/Nrf-2/HO-1 inhibited NF-κB/COX-2/iNOS pathways to decrease pro-inflammatory cytokines (IL-1β, IL-6, and TNF-α)”. Reconstruct the statement. Too long and confusing.
-Line 485-487; “The further identified bioactive ingredients components and proposals of possible mechanisms to obtain those bioactive compounds in the 486 molecular mechanism will continue the further investigation.”. Statements Not necessary.
-line 488-489; “function of food or 488 nutraceutical pharmacy," reframe the sentence.
English Language:
Revisions in English grammar and tense corrections have to be made. I recommend the authors proofread the final draft with an English-speaking scientific colleague or use an English-speaking scientific editing service to reduce grammar and tense errors.
Author Response
Dear Reviewer,
Thank you for your comments and suggestions. Please find the detailed response below and the corresponding corrections highlighted in the resubmitted files.
Sincerely yours,
Tuzz-Ying Song, PhD., Professor.
